# Attenuation of Avian Flavivirus by Rewiring the Leucine and Serine Codons of Its E-NS1 Protein toward Stop Mutation To Redirect Virus Evolution

Jiaqi Guo,[a] Yu He,[a] Xiaoli Wang,[a] Andres Merits,[d] Mingshu Wang,[a,b,c] Renyong Jia,[a,b,c] Dekang Zhu,[b,c] Mafeng Liu,[a,b,c] Xinxin Zhao,[a,b,c] Qiao Yang,[a,b,c] Ying Wu,[a,b,c] Shaqiu Zhang,[a,b,c] Juan Huang,[a,b,c] Sai Mao,[a,b,c] Xumin Ou,[a,b,c] Qun Gao,[a,b,c] Di Sun,[a,b,c] Bin Tian,[a,b,c] Anchun Cheng,[a,b,c] Shun Chen[a,b,c]

[a]Institute of Preventive Veterinary Medicine, Sichuan Agricultural University, Chengdu, Sichuan, China

[b]Research Center of Avian Disease, College of Veterinary Medicine, Sichuan Agricultural University, Chengdu, Sichuan, China

[c]Key Laboratory of Animal Disease and Human Health of Sichuan Province, Sichuan Agricultural University, Chengdu, Sichuan, China

[d]Institute of Technology, University of Tartu, Tartu, Estonia

Jiaqi Guo, Yu He, and Xiaoli Wang contributed equally to this work. Author order was determined on the basis of their contributions.

**ABSTRACT** Recently, a new strategy for attenuating RNA viruses by redirecting their evolution in sequence space was confirmed for *Enterovirus* and *Influenza* viruses. Using avian flavivirus as a model, the 69 serine and 53 leucine codons on the E-NS1 genes were modified to change evolutionary direction of the viral sequence space. This means that all codons encoding serine or leucine residues were substituted with codons that are only one base different from the three stop codons, resulting in the initial position of the virus genome in sequence space being closer to the detrimental areas to achieve attenuation by reducing viral adaptability. The growth curve and plaque size of CQW1-one-to-stop (CQW1-OTS) were similar to those of CQW1-wild type (CQW1-WT) *in vitro*, but attenuated proliferation was detected when treated with a mutagenic reagent (ribavirin). However, comparably high CQW1-OTS and CQW1-WT lethality rates were detected in 9-day-old duck embryos and 5-day-old ducklings, suggesting that this strategy works but with limitations. With that in mind, homologous hosts in nonsensitive age (25-day-old ducks) and heterologous hosts (3-week-old Kunming mice) were employed to investigate if CQW1-OTS was attenuated under host selection pressure. Minimal attenuation of CQW1-OTS in elder ducks and apparent attenuation in mice were reported, providing reduced viral titers, mild clinical signs, and lower specific infectivity. Collectively, we experimentally demonstrate that the attenuation strategy of redirecting virus evolution in sequence space works for flavivirus. Redirection of the virus is attenuated only under some outside pressure, such as heterologous hosts or antiviral drugs treatment, limiting its usage in flaviviruses.

**IMPORTANCE** Flaviviruses are medically important arboviruses that threaten public health, but no approved treatments are currently available. Vaccines prevent flavivirus infection. We employed duck Tembusu virus (TMUV), a mosquito-borne flavivirus, to evaluate virus redirection. TMUV is native to birds and could infect mice by intracerebral injection, making it an experimental animal model to study flavivirus characteristics *in vivo*. The 69 serine and 53 leucine codons on the E-NS1 proteins of CQW1 were synonymously substituted to change evolutionary direction of the virus in sequence space. *In vitro* mutagen reagent treatment suppressed CQW1-OTS viral multiplication, but *in vivo* attenuation depended on host selective pressure. CQW1-OTS viral attenuation was observed in older ducks but not sensitive ducklings; considerable attenuation was also observed in heterogenous host (mice), which provides more selective pressure on viruses. Collectively, these data indicated that there are very important preconditions for application of evaluating whether this strategy shows application prospects in novel flavivirus vaccine development.

Address correspondence to Anchun Cheng, chenganchun@vip.163.com, or Shun Chen, shunchen@sicau.edu.cn.

The authors declare no conflict of interest.

**KEYWORDS** duck Tembusu virus, attenuated strategy, sequence space, selective pressure

Flavivirus, an arthropod-borne virus, is a major threat to global public health. To date, there is no specific treatment for these virus-related diseases, and vaccination is the most important means of effective prevention and control. Live attenuated vaccines (LAVs) represent a direction in the development of flavivirus vaccines, such as live attenuated 17D vaccine strains against YF virus (1, 2) and vaccine strain SA14-14-2 against JEV (3, 4); however, most of these attenuated flavivirus strains were naturally attenuated by serial passages. Moratorio et al. proposed that RNA viruses can be rationally attenuated by redirecting their evolutionary trajectories toward detrimental areas of their sequence space (5). Simply put, the viral genome defines the location of a virus population in sequence space, and the position will be changed when its bases are adjusted accordingly. Different positions also mean the different mutational neighborhoods that the viral genome faces, and the quality of the mutational neighborhoods can determine the evolutionary potential (applicability) of the viral genome. The evolutionary potential of a virus in turn determines its ability to withstand evolutionary pressures. Therefore, in theory, the evolutionary potential of a modified virus is greatly reduced under the conditions of this strategy, and it is straightforward to produce lethal mutations (termination codons) to attenuate the virus under the pressure of the environment. In theory, this strategy can be applied to all RNA viruses, the operation is simple and fast, and the attenuation effect is remarkable.

Many flaviviruses that are pathogenic to humans are relatively well studied. However, due to a lack of adequate *in vivo* models, multiple vital aspects of flavivirus infection are poorly understood. Duck Tembusu virus (DTMUV) is classified into the *Flavivirus* genus and the *Flaviviridae* family (6). It is a mosquito-borne single-stranded positive-sense RNA virus. The viral genome is approximately 11,000 nucleotides (nt) in length and encodes one open reading frame (ORF) containing three structural proteins (C, prM, and E) and seven nonstructural proteins (NS1, NS2A, NS2B, NS3, NS4B, and NS5) (7, 8). TMUV is native to birds and can also infect mice by intracranial (IC) injection (9), providing an excellent animal model to study flaviviral characteristics *in vivo*. Here, to take advantage of this, TMUV was used as a representative flavivirus model to investigate whether the redirected viral evolutionary strategy can be applied to the rapid development of flavivirus LAVs. Fifty-three leucine and 69 serine on the E protein and NS1 protein of TMUV (distinguishing the effect of codon deoptimization, only 5% modification) were rationally designed. The growth characteristics and virulence of the engineered recombinant virus were assessed both *in vitro* and *in vivo*.

## RESULTS

**Recovery and identification of CQW1-OTS.** The schematic diagram for CQW1-one-to-stop (CQW1-OTS) is shown in Fig. 1A. Cytopathic effect (CPE) (Fig. 1B) and immunofluorescence assay (IFA) (Fig. 1C) at 72 days postinfection (dpi) showed that CQW1-OTS was recovered successfully. The supernatant was collected to infect fresh BHK-21 cells for 10 passages. Passages 1, 3, 5, 7, and 10 were sequenced, and no unexpected mutations were observed. The F1 generation virus was used in all subsequent experiments.

Next, we further assessed the fitness of CQW1-OTS *in vitro* under certain pressures. Viruses were treated with different concentrations of ribavirin or 5-fluorouracil (5-FU) (base analogs). Under the first round of treatment with the mutagenic agent (Fig. 1D), CQW1-OTS presented significantly lower fitness than CQW1-wild type (CQW1-WT) (100 $\mu$M ribavirin, 200 $\mu$M 5-FU, and 400 $\mu$M 5-FU). Ribavirin was selected for five continuous rounds of pressurization to verify the properties of CQW1-OTS under ongoing high pressure. Compared with CQW1-WT, the decreased fold changes in viral titers of CQW1-OTS increased from 9.6 (Fig. 1D) to 3,086.3 (Fig. 1E) at a concentration of 100 $\mu$M after 5 rounds of treatment. This may be because the virus has been engineered to relocate in sequence space, increasing the probability of nonsense mutations. Taken together, these data demonstrated that CQW1-OTS was successfully attenuated *in vitro* under ribavirin treatment.

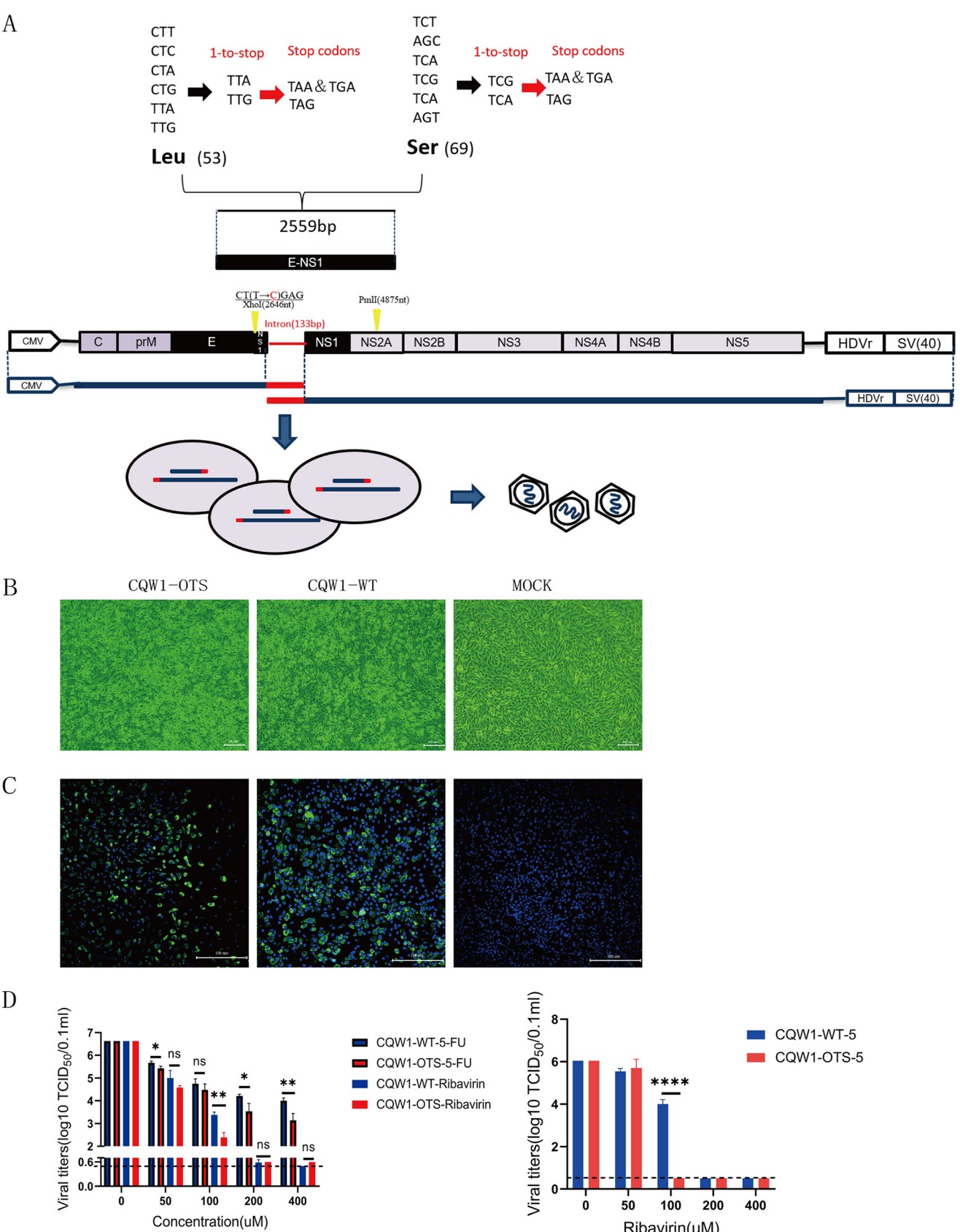

**FIG 1** Recovery and identification of CQW1-OTS. (A) Schematic diagram for generating CQW1-OTS. (B) CPE caused by CQW1-OTS. At 72 h posttransfection, clear CPE was observed in BHK-21 cells transfected with CQW1-OTS amplicons. (C) Recovery of CQW1-OTS confirmed by IFA and

**CQW1-OTS was comparable to CQW1-WT *in vitro* and *in vivo*.** To verify the *in vitro* and *in vivo* properties of successfully attenuated viruses, we first assessed the growth kinetics, plaque morphology, and virulence (in duck embryos) of CQW1-OTS and CQW1-WT *in vitro*. As shown in Fig. 2A, the viral growth kinetics of CQW1-OTS are similar to those of CQW1-WT in both mammalian BHK-21 and avian duck embryo fibroblast (DEF) cells, and both viruses produced uniform plaque morphology and were approximately the same size on BHK-21 cells (Fig. 2B). We further compared the virulence of CQW1-OTS and CQW1-WT in 9-day-old duck embryos (Fig. 2C). The early death of embryos in both infected groups was observed at 4 dpi, and the mortality of both the CQW1-OTS and CQW1-WT groups was 50% within 10 dpi; the statistical analysis result was nonsignificant. Overall, these results indicate that CQW1-OTS was not attenuated *in vitro*.

Then, 5-day-old ducklings were infected with CQW1-OTS, CQW1-WT, or Dulbecco modified Eagle medium (DMEM) to verify whether CQW1-OTS is attenuated in its natural host. The experimental design for the animal experiment is shown in Fig. 2D. No significant differences in weight change were observed between these two virus-infected groups, and both were remarkably lower than the DMEM group (Fig. 2E). The CQW1-OTS and CQW1-WT groups developed obvious viremia with similar titers (Fig. 2F). At 4, 5, 6, and 7 dpi, one, three, four, and one ducklings died in the CQW1-WT group (90%), respectively; much the same phenomenon occurred in the CQW1-OTS group, and 80% of ducklings died (six, one, and one ducklings died on days 5, 6, 8 after infection, respectively) (Fig. 2G). Most of the infected ducklings gradually lost appetite and were reluctant to move; then, unstable standing and paralysis of the legs were observed, and they finally died. The remaining ducklings gradually recovered after 7 dpi (Fig. 2H). These two viruses can cause equally high mortality in the infected ducklings. These results indicated that CQW1-OTS virus failed to attenuate in the sensitive nature host (duckling). This leads us to hypothesize that there may be some requirements for this attenuated method, such as a certain selective pressure.

**CQW1-OTS was mildly attenuated in 25-day-old ducks.** The inconsistent pathogenicity with duck TMUV infections in ducks was related to the duck's age at infection (10–12). The ducks are more resistant to TMUV as they age. Consequently, 25-day-old ducks were selected as the animal model to further assess the properties of CQW1-OTS *in vivo*. One duck that was intramuscularly infected with CQW1-WT died at 7 dpi (Fig. 3A). There was no noticeable difference in viral tissue load levels between CQW1-WT and CQW1-OTS in the liver, spleen, and brain in the early infection phase (3 dpi). However, significantly lower viral loads were detected in CQW1-OTS-infected ducks at 5 dpi (Fig. 3B) than at 3 dpi, while viral loads in the CQW1-WT-infected groups further increased to higher levels at 5 dpi, suggesting that the proliferation of CQW1-OTS may be attenuated and cleared faster by the host than by the WT virus. This also explains why the more robust innate immune responses were stimulated by CQW1-WT at 5 dpi and induced significantly higher levels of alpha interferon (IFN-$\alpha$) and IFN-$\beta$ (Fig. 3C). At 14 dpi, a remarkably enhanced lymphocyte proliferation response was observed in the CQW1-WT group (Fig. 3D). However, at 14 dpi, IFN-$\gamma$ and interleukin-4 (IL-4) in duck serum samples (Fig. 3E) were equivalent between both groups, and detectable neutralizing antibodies induced by CQW1-WT and CQW1-OTS viruses at all tested time points (from 1 to 7 weeks) showed no significant difference (Fig. 3F). This means that mutant viruses may have immunogenicity similar to that of CQW1-WT. Compared with 5-day-old duckling data, mild attenuation of CQW1-OTS was confirmed in ducks of nonsensitive age.

**CQW1-OTS only showed lower fitness in a heterologous host.** According to the above finding, we wondered if CQW1-OTS would be attenuated as expected under greater evolutionary pressure. In this case, the relative fitness of CQW1-OTS and

**FIG 1** Legend (Continued)
mouse anti-TMUV polyclonal antibody used as the primary antibody. (D) Sensitivity of CQW1-OTS and CQW1-WT to ribavirin and 5-FU. (E) The sensitivity of CQW1-OTS and CQW1-WT to ribavirin after five consecutive cycles of treatment. The graphs show the mean and standard error of the mean (SEM); $n = 3$. Statistical significance was determined using an unpaired $t$ test; ns, not significant; *, $P < 0.05$; **, $P < 0.01$; ****, $P < 0.0001$.

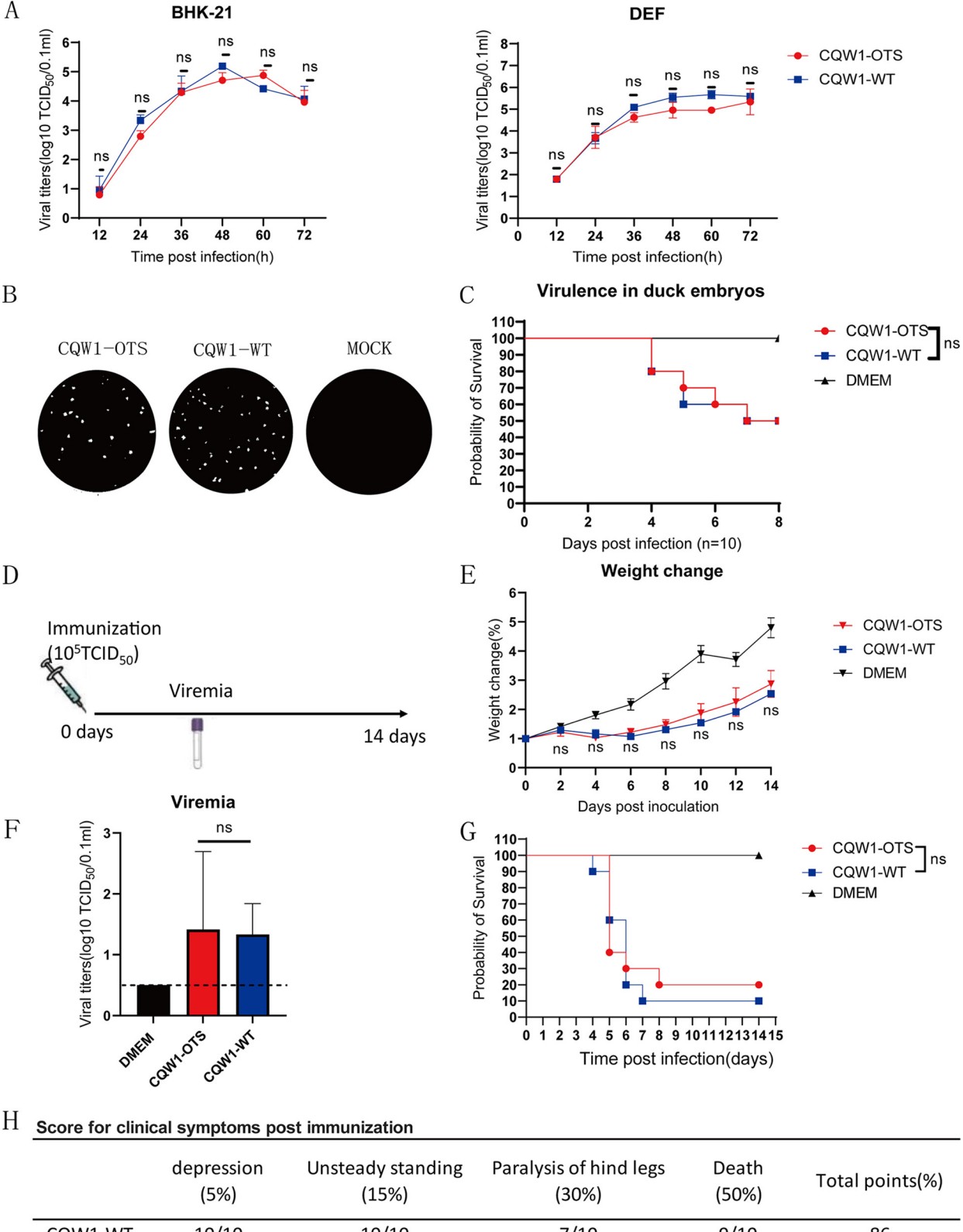

**FIG 2** Characteristics of CQW1-OTS *in vitro* and *in vivo*. (A) Growth curve of CQW1-OTS and CQW1-WT on BHK-21 and DEF cells. Statistical significance was determined using multiple *t* test; ns, not significant. (B) Plaque morphology of CQW1-OTS and CQW1-WT on BHK-21 cells at

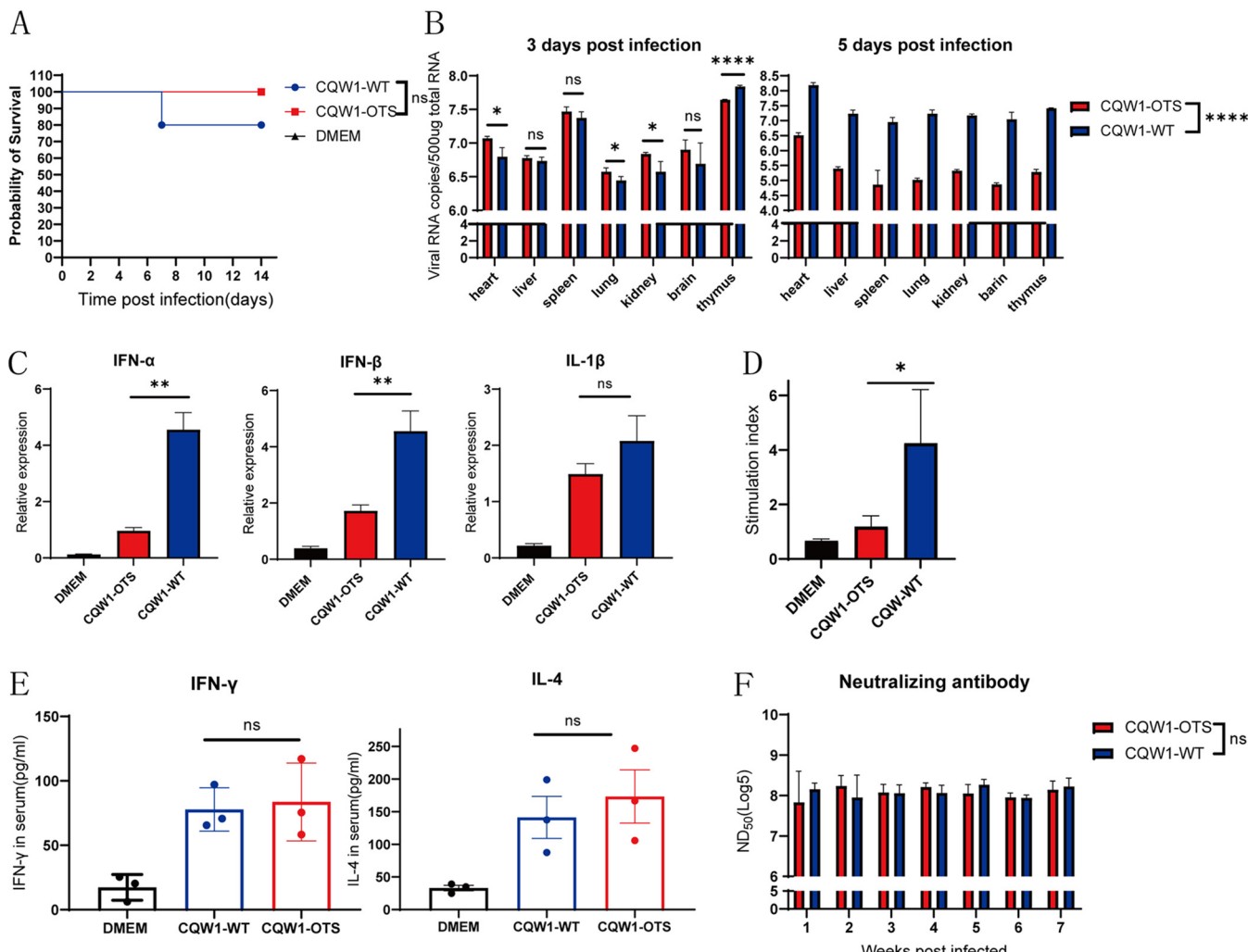

**FIG 3** Immunogenicity characteristics of CQW1-OTS in 25-day-old ducks. (A) Survival rate of duck postinfection. The statistical significance of survival was analyzed using a survival curve and the log-rank (Mantel-Cox) test, with significance defined. Ns, not significant. $n = 5$. (B) Viral copies (at 3 and 5 dpi) in the heart, liver, spleen, lung, kidney, brain, and thymus of 25-day-old ducks were detected by RT-qPCR. (C) Relative mRNA expression of IFN-$\alpha$, IFN-$\beta$, and IL-1$\beta$ in duck spleen at 5 dpi. (D) Peripheral T-lymphocyte (from infected ducks) proliferation activated by TMUV infection. (E) Serum (14 dpi) levels of IFN-$\gamma$ and IL-4 were determined using ELISA. (F) Neutralization antibody levels in the serum were determined with PRNT. Statistical significance was determined using an unpaired $t$ test. Ns, not significant; *, $P < 0.05$; **, $P < 0.01$; ****, $P < 0.0001$.

CQW1-WT was measured in a heterologous host—3-week-old mice. Body weight started to decrease from 4 dpi (CQW1-WT) and 5 dpi (CQW1-OTS) compared with the control group and increased again until 8 dpi (Fig. 4A). Four mice in the CQW1-WT group died at 5, 7, 10, and 11 days. Only 20% of the mice died from CQW1-OTS infection (two mice died on days 8 and 9) (Fig. 4B). Notably, although no obvious difference in body weight change was observed, high viral titers were detected in the brains of the CQW1-WT group, especially at 2 dpi and 8 dpi (Fig. 4C). The CQW1-WT virus-infected mice exhibited more severe clinical symptoms than the other mice, including

**FIG 2** Legend (Continued)

120 h. (C) Virulence of CQW1-OTS and CQW1-WT in 9-day-old duck embryos ($n = 10$). Each embryo was infected with 100 $\mu$L of CQW1-OTS or CQW1-WT at a dose of $10^3$ TCID$_{50}$. The log-rank (Mantel-Cox) survival analysis test was performed for statistical significance; ns, not significant. (D) Experimental design for animal experiments. Each duckling was infected with 200 $\mu$L of CQW1-OTS or CQW1-WT at a dose of $10^5$ TCID$_{50}$/0.1 mL. Blood was collected at 3 dpi for viremia assays. (E) Weight changes of CQW1-OTS- and CQW1-WT-infected ducklings. Statistical significance was determined using a multiple $t$ test; ns, not significant. (F) Viremia caused by CQW1-OTS and CQW1-WT infection. Statistical significance was determined using an unpaired $t$ test; ns, not significant. (G) Survival rate of 5-day-old ducklings postchallenge. The statistical significance of survival was analyzed using a survival curve and the log-rank (Mantel-Cox) test, with significance defined; ns, not significant. (H) Clinical symptoms of CQW1-OTS- and CQW1-WT-infected ducklings. In each item, one score corresponding to one duck appeared as a symptom. The total points = (5% × item 1 + 15% × item 2 + 30% × item 3 + 50% × item 4) × 100%.

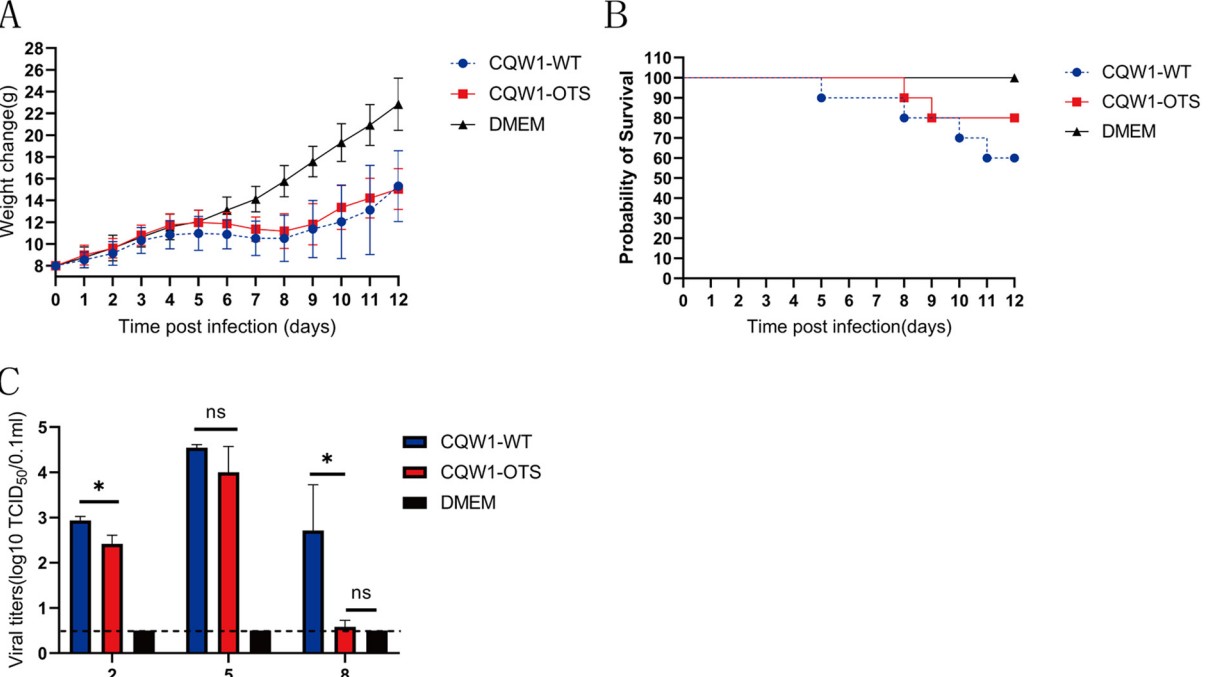

### D

**Score for clinical symptoms post immunization**

|  | depression 10% | Paralysis 30% | Blindness 15% | death 45% | Total score(%) |
|---|---|---|---|---|---|
| CQW1-WT | 10/10 | 2/10 | 4/10 | 4/10 | 40 |
| CQW1-OTS | 2/10 | 3/10 | 2/10 | 2/10 | 23 |
| DMEM | 0 | 0 | 0 | 0 | 0 |

### E

**Differences in specific infectivity between mouse brain rTMUV and pTMUV**

| days post infected | Virus | Titer ($\log_{10}TCID_{50}/0.1ml$) | $VC^a$ ($\log10VC$) | Specific infectivity ($VC/TCID_{50}$) | Relative specific infectivity |
|---|---|---|---|---|---|
| 2 | CQW1-OTS | 2.417 | 5.341 | 2.21 | 1 |
|  | CQW1-WT | 3.25 | 5.758 | 1.772 | 0.802 |
| 5 | CQW1-OTS | 4 | 6.075 | 1.519 | 1 |
|  | CQW1-WT | 4.542 | 6.613 | 1.456 | 0.959 |
| 8 | CQW1-OTS | 0.583 | 5.41 | 9.273 | 1 |
|  | CQW1-WT | 2.708 | 6.138 | 2.266 | 0.244 |

[a] Viral RNA copies/500ug total RNA

**FIG 4** *In vivo* properties of CQW1-OTS in 3-week-old Kunming mice. Mice were intracerebrally inoculated with 50 $\mu$L of CQW1-WT or CQW1-OTS viruses ($10^{5.25}$ TCID$_{50}$/0.1 mL). (A) Weight change of CQW1-OTS- and CQW1-WT-infected mice. (B) Survival rate of mice infected with CQW1-OTS and CQW1-WT. (C) Viral loads in the brains of mice, determined by TCID$_{50}$. Statistical significance was determined using an unpaired *t* test; ns, not significant; *, $P < 0.05$. (D) Clinical symptoms postinfection. Total points = 10% × item 1 + 30% × item 2 + 15% × item 3 + 45% × item 4. (E) Specific infectivity of the virus in brain tissues.

depression, anorexia, paralysis of the hind legs, and blepharitis (Fig. 4D). Mice inoculated with CQW1-WT virus had a poor prognosis, and blindness or paralysis continued to the end of the trial. Meanwhile, the relative specific infectivity (Fig. 4E) of CQW1-OTS was significantly higher than that of CQW1-WT at 8 dpi, which means that CQW1-OTS required a higher number of particles to achieve the same level of infection as the WT virus. Collectively, under high host selective pressure, the adaptability of CQW1-OTS is lower than that of CQW1-WT.

## DISCUSSION

RNA viruses lack proof-reading mechanisms, and high replication rates among most make them inherently error prone. The lack of 3′–5′ exonuclease activity in its RNA polymerase causes high mutation rates during viral genome replication (13). Although the majority of mutations are deleterious, the high mutation rate will produce potential beneficial mutations at the population level, helping the virus evade selection pressure (14). RNA viruses can have highly diverse populations with few replication cycles (15, 16), which is considered the basis for improving their adaptability to the environment. The emergence of new variants will be able to resist antiviral approaches, escape immune responses, alter tissue tropism, or cross species barriers (17). TMUV belongs to the flaviviruses and was first identified in *Culex* mosquitoes in Malaysia in 1955 (18). As a mosquito-borne *flavivirus*, it has the characteristic of an arbovirus obligate cycle between vertebrates and arthropods, and this process increases fitness after alternation (14, 19). In 2000, a broiler farm located in Malaysia first reported an infection of TMUV in avians (Sitiawan) (20). In 2010, TMUV caused the first outbreaks in egg-laying ducks in China, and ever since then, TMUV has begun to proliferate in many countries in Asia (6, 9). In this study, the TMUV strain that we used was CQW1 (GenBank accession number KM233707.1), which was isolated early from diseased ducks (21). Accordingly, a full-length infectious cDNA clone, noninfectious replicon, and reporter DTMUV were constructed by our lab (22–24). These molecular tools provide a good platform to study the designed attenuation of TMUV. For best application, the leucine and serine codons of the E-NS1 protein were rewired to constrain the expansion of the viral populations toward detrimental mutational neighborhoods (stop mutations). The CQW1-OTS engineered by this strategy was more sensitive to high selection pressure from ribavirin and 5-FU (two mutagenic agents) than CQW1-WT. This also demonstrates that the adaptability of the modified CQW1-OTS is much lower than that of the CQW1-WT. In other words, the strategy was suitable for TMUV, and the modified CQW1-OTS virus was successfully attenuated under certain conditions (high pressure) *in vitro*. However, CQW1-OTS in this study failed to attenuate in 5-day-old ducklings, and slight attenuation occurred in older young ducks but showed significant differences in mice. This implies that the number of lethal mutations in CQW1-OTS is insufficient under conditions of insufficient selection pressure. This method was first reported in *Nature Microbiology* in 2017 (5), and the attenuation strategy was suitable for vaccine development of RNA viruses, which have been used in wild-type influenza A virus—A/Paris/2590/2009 (H1N1pdm09) (25, 26)—and wild-type Coxsackie virus B3 (Nancy strain) (27, 28). In that study, both of Influenza A virus and coxsackievirus B3 natural hosts were human, which may be the reason that this attenuation strategy worked well in the heterologous host (5-week-old BALB/c male mice). In general, we speculate that the attenuation mechanism of CQW1-OTS was based on pressure screening and that the increase in lethal mutations during the evolution of viruses leads to attenuation.

Interestingly, the same strategy was successfully used in porcine circovirus type 2 (PCV2) (GenBank accession number KR816332) (29). PCV2 can cause postweaning multisystemic wasting syndrome (PMWS) by infecting the immune system of animals and preventing the organism from producing a normal immunological response (30, 31). However, its clinical symptoms and pathological damage are atypical and mild (32). The most common cause of mortality is secondary infection. PCV2 and CQW1-OTS recombinant viruses produced comparable outcomes in natural hosts, inducing a lower level of interferon, a higher level of neutralizing antibodies, and a significantly

lower viral tissue load (lesion score). The PCV2 mutant strain exhibited comparable results to its vaccine control group, demonstrating that it could be employed as an alternative vaccine. Although no vaccination group was established in this experiment, CQW1-OTS was able to induce equivalent amounts of neutralizing antibodies, INF-$\gamma$, and IL-4 as CQW1-WT. In addition, ducklings that survived their initial infection with CQW1-OTS were immune to reinfection by the highly pathogenic wild virus. Therefore, if we eliminate the difference in DTMUV and PCV2 lethality to their natural hosts, the two findings do not contradict one another. A limitation of this study was that deep sequencing was not performed in the animals to determine the mutation frequency of the stop codon. Of course, we do not rule out the possibility that continuing to increase the length of the sequence modification will affect the degree of attenuation. However, when the proportion of modified codons exceeds 5%, the reason for its attenuation may be codon deoptimization. Therefore, under the current experimental conditions, the conclusion applies. Although this study failed to attenuate TMUV *in vivo*, we demonstrated that the attenuation strategy in this study is useful and supplemented the applicable conditions: the attenuation mechanism of CQW1-OTS is based on pressure screening. This strategy may not be directly implemented as an LAV strategy against RNA viruses.

## MATERIALS AND METHODS

**Ethics statement.** All animal experimental procedures were approved by the Institutional Animal Care and Use Committee of Sichuan Agriculture University in Sichuan, China—protocol permit number SYXK(川) 2019-187.

**Viruses, cells, chemicals, and animals.** Baby hamster kidney (BHK-21) cells were maintained in modified Eagle's medium (DMEM) (Gibco, Shanghai, China) with 5% fetal bovine serum (FBS) (Gibco, New York, USA) and incubated at 37°C with 5% $CO_2$. Duck embryo fibroblast (DEF) cells were harvested from 9-day-old duck embryos, cultured in DMEM supplemented with 10% newborn calf serum (NBCS) (Gibco, New York, USA) and incubated at 37°C with 5% $CO_2$. The early TMUV CQW1-WT (GenBank accession number KM233707.1) was rescued using the reverse genetic method reported previously by our lab (22, 23). The epidemic strain TMUV CHN-YC (GenBank accession number MN966680.1) was isolated in 2019 (a gift from Rui Luo, Huazhong Agricultural University). 5-Fluorouracil and ribavirin were purchased from MCE (Shanghai, China) and solubilized in DMEM with 2% FBS. Specific-pathogen-free (SPF) ducklings were purchased from the National Laboratory Poultry Animal Resource Center (Harbin Veterinary Research Institute, China). Twenty-five-day-old ducks and duck embryos were purchased from the Waterfowl Breeding Center of Sichuan Agriculture University. Three-week-old Kunming mice were purchased from Chengdu DOSSY Experimental Animals Co., Ltd. (Chengdu, China).

**Recovery of CQW1-OTS by infectious subgenomic amplicons.** The recombinant TMUV with codon-modified E-NS1 gene (CQW1-OTS) was generated with the infectious subgenomic amplicons method (33). The design scheme is shown in Fig. 1A. The codon-modified E-NS1 sequence was obtained by whole-gene synthesis. To simplify the assembly, a silent mutation (T→C) was induced in position 2646 by primers to generate a unique restriction site XhoI. The codons encoding leucine (CTT, CTC, CTA, CTG, TTA, TTG) and serine (TCT, AGC, TCA, TCG, TCA, AGT) in the E-NS1 gene were modified as TTA, TTG or TCG, TCA, respectively. The modified codon differs from one of the three stop codons (TAA, TGA, or TAG) by only a single nucleotide. To construct CQW1-OTS, we utilized the TMUV full-length cDNA infectious clone pACNR-CQW1-Intron as the template (24). Two cDNA fragments covering the entire genome of TMUV with lengths of approximately 3,307 bp and 8,663 bp, which took the intron (133 bp) as the overlapping region, were obtained using PCR.

To rescue CQW1-OTS, BHK-21 cells were seeded in 12-well plates. After 16 h of attachment with 70 to 90% confluence, the cells were transfected with an equimolar mixture of the two fragments obtained as mentioned above for a total final amount of 1 $\mu$g per well using Lipofectamine 3000 (Thermo Fisher Scientific, Shanghai, China) according to the manufacturer's instructions, while the mock group received the same dose of transfection reagent. Six hours later, the supernatant was removed and replaced with 1 mL DMEM containing 2% FBS and incubated at 37°C with 5% $CO_2$. When a cytopathic effect (CPE) appeared, the supernatant was harvested (F0 virus) and used again to infect fresh BHK-21 cells. The supernatant was harvested when 70% to 80% CPE appeared, clarified by centrifugation, and stored at −80°C for use in performing whole-genome sequencing and subsequent experiments.

**Indirect immunofluorescence assay.** IFA was performed as previously reported (24). In brief, cells were washed with phosphate-buffered saline (PBS) three times, fixed with 4% paraformaldehyde for 1 h at 4°C, and permeabilized for 1 h at 4°C with 0.3% Triton in PBS. After 1 h of incubation in a blocking buffer containing 5% bull serum albumin (BSA) in PBS, the cells were treated with mouse anti-TMUV polyclonal antibody (self-prepared using inactivated TMUV virion, which recognizes E and M proteins using Western blotting assay confirmation) at a dilution of 1:300 with 1% BSA in PBS for 2 h and then incubated with goat anti-mouse IgG conjugated with Alexa Fluor 488 (Thermo Fisher Scientific, Shanghai, China) for 1 h at 37°C. Finally, the cells were stained with 4′,6-diamidino-2-phenylindole (DAPI) in PBS for 15 min. The cells were washed three times with 1% PBS/Tween 20 (PBST) after each step. Finally, fluorescence images were acquired under a fluorescence microscope (Nikon, Tokyo, Japan).

**Virus titration and plaque assay.** Viral titers were determined using the median tissue culture infection dose ($TCID_{50}$) as previously reported (24). Tenfold serial dilutions of the virus samples were prepared in serum-free DMEM ($10^{-1}$ to $10^{-10}$) with 1% penicillin/streptomycin, and 100 $\mu$L dilutions of the viral sample were distributed to each of eight wells of a 96-well plate containing BHK-21 cells. After 5 days of incubation, the absence of CPE in each well was determined by microscopy. Viral titers were calculated according to the Karber method.

For the plaque assay, CQW1-WT or CQW1-OTS were 10-fold serially diluted in DMEM; 400-$\mu$L virus samples of each dilution were added to BHK-21 cells at approximately 95% confluence in a 12-well plate. After 1.5 h of attachment at 37°C, 1 mL of 1% methyl cellulose overlay containing 2% FBS was added to each well, and the plate was incubated for 5 to 7 days. Later, the overlay was removed, and the plate was washed three times with PBS, fixed with 4% formaldehyde at room temperature for 20 min, and then stained with 1% crystal violet for 1 min. Finally, the cells were washed carefully, and visible plaques were observed.

**Growth curve.** Separately, DEF and BHK-21 cells were seeded in 12-well plates and cultured with 5% $CO_2$ at 37°C. The culture medium was removed, and the cells were washed three times with PBS when the cells were approximately 80% confluent, infected with the virus at 100 $TCID_{50}$, and incubated at 37°C. After allowing viral adsorption to proceed for 90 min (with 5% $CO_2$ at 37°C), the inoculum was removed. The cell monolayers were washed twice with PBS. Each well was then filled with 1 mL DMEM containing 2% FBS and 1% penicillin/streptomycin. The plates were incubated for 72 h; every 12 h, 500 $\mu$L of the supernatant was collected and subjected to viral titration as described above.

**Virulence of CQW1-OTS in duck embryos.** Thirty 9-day-old duck embryos were randomly divided into three groups and injected with 100 $\mu$L ($10^3$ $TCID_{50}$/0.1 mL) of CQW1-OTS or parental TMUV (CQW1-WT) by allantoic cavity inoculation. DMEM was used to dilute the virus stocks to the desired concentration. Embryonic eggs were incubated at 37°C and candled daily. If the embryos lost movement and blood vessels were desquamated, the embryos were regarded as dead. The duration of survival of inoculated eggs was recorded every day.

**Genetic stability of viruses.** To evaluate the genetic stability of CQW1-OTS, the virus was serially passaged 10 times in BHK-21 cells, and the 1st, 3rd, 5th, 7th, and 10th passages were sequenced to determine its genetic stability.

**Mutagenic reagent treatments.** BHK-21 cell monolayers in 12-well plates were infected with CQW1-OTS or CQW1-WT (200 $TCID_{50}$) at 37°C with 5% $CO_2$ for 1 h. The virus-compound mixture was removed, and the cells were washed twice with PBS. Then 1 mL ribavirin or 5-FU at varying concentrations (0, 50 $\mu$M, 100 $\mu$M, 200 $\mu$M, 400 $\mu$M) was added to each well. At 48 h postinfection, viruses were collected in clarified supernatant to determine virus titers based on the $TCID_{50}$ method. The same ribavirin procedure was repeated for five passages under each distinct mutagenic condition in three biological replicates.

**TMUV infections *in vivo*.** To test the virulence of CQW1-OTS in ducklings. Groups of 10 SPF 5-day-old ducklings were injected intramuscularly (IM) with 200 $\mu$L $10^5$ $TCID_{50}$/0.1 mL of CQW1-OTS or CQW1-WT. Ducklings in the mock group were incubated in the same route and volume of DMEM. The ducklings were weighed every 2 days, and survival and clinical signs were monitored every day. Blood was collected from ducklings at 3 days postinfection (dpi) for viremia assays.

To evaluate the virulence and serological responses against TMUV stimulation in ducks, fifteen 25-day-old ducks were randomly divided into three groups, and each group received an infected intramuscular injection with 200 $\mu$L $10^5$ $TCID_{50}$/0.1 mL of CQW1-OTS or DMEM as a mock control. Serum samples were collected from all groups weekly for 7 weeks to monitor the neutralization antibodies. Simultaneously, ducks were bled for the T-lymphocyte proliferation assay, and the levels of IFN-$\gamma$ and IL-4 in serum were also analyzed using enzyme-linked immunosorbent assay (ELISA) at 14 dpi. To determine the number of viruses sustained in organs infected with CQW1-OTS, three groups of ducks (six per group) were intramuscularly inoculated with CQW1-OTS. Each group of three ducks was humanely euthanized at 3 and 5 dpi, and their organs, including the heart, liver, spleen, lung, kidney, brain, and thymus, were obtained to determine virus titers via reverse transcriptase quantitative PCR (RT-qPCR) assay.

To estimate the virulence of CQW1-OTS in mice, thirty 3-week-old mice (female) were randomly divided into three groups ($n = 10$) and infected with 50 $\mu$L of CQW1-WT, CQW1-OTS ($10^{5.25}$ $TCID_{50}$/0.1 mL), or an equal volume of DMEM via IC injection. The mice's weight, survival, and clinical signs were checked every day. Each group of three mice was humanely euthanized at 3, 5, and 8 dpi, and their brains were obtained to determine virus titers by RT-qPCR assay and $TCID_{50}$ in BHK-21 cells.

**Detection of viremia in the infected ducks.** The ducks were bled utilizing a sterile needle and syringe. The blood samples in a serum tube were incubated at 37°C for 2 h and then centrifuged for 10 min (at approximately 500 g) to separate the serum from the red blood cells. A serially diluted (1:10) serum and medium mixture was added to freshly seeded BHK-21 cells and cultured for 5 days. Titers were determined according to the Karber method.

**Lymphocyte proliferation assay.** Lymphocytes were separated from blood using a peripheral blood lymphocyte separation kit (Solarbio, Beijing, China) following the manufacturer's recommendations. Eighty microliters of cell-containing RPMI 1640 medium (Gibco, Shanghai, China) containing 10% FBS and supplemented with 1% penicillin/streptomycin was transferred to 96-well plates after cell counting. Twenty microliters of epidemic CHN-YC virus ($10^{6.25}$ $TCID_{50}$/0.1 mL) was added to specifically stimulate the proliferation of lymphocytes. The same volume medium was added to the mock groups. Cell proliferation was detected after cells were cultured at 37°C with 5% $CO_2$ for 36 h using the cell counting kit 8 (MCE, Shanghai, China) following the manufacturer's instructions.

**Detection of IFN-$\gamma$ and IL-4 in serum by ELISA.** Th1-type cytokine IFN-$\gamma$ and Th2-type cytokine IL-4 in serum at 14 dpi were measured using commercial duck IFN-g and IL-4 sandwich ELISA kits (MLbio, Shanghai, China) per the manufacturer's instructions.

**TABLE 1** Primer sequences

| Gene name | Primer sequences (5′ to 3′) |
| --- | --- |
| Duck IFN$\alpha$ | F: TTGCTCCTTCCCGGACA |
| | R: GCTGAGGGTGTCGAAGAGGT |
| Duck IFNB1 | F: TCTACAGAGCCTTGCCTGCAT |
| | R: TGTCGGTGTCCAAAAGGATGT |
| Duck IL-1$\beta$ | F: TCGACATCAACCAGAAGTGC |
| | R: GAGCTTGTAGCCCTTGATGC |
| Duck $\beta$-actin | F: GATCACAGCCCTGGCACC |
| | R: CGGATTCATCATACTCCTGCTT |
| TMUV-E | F: AATGGCTGTGGCTTGTTTGG |
| | R: GGGCGTTATCACGAATCTA |

**Plaque reduction neutralization test.** BHK-21 cell cultures were prepared in 12-well plates as described above. The cells were washed with PBS twice and inoculated with 0.2 mL of the virus-serum mixtures when confluent. Serum (inactivated at 56°C for 30 min) was diluted in a 5-fold step to $5^{-7}$ with serum-free DMEM. Each sample was mixed with an equal volume of 120 TCID$_{50}$ CQW1-WT or CQW1-OTS virus; the same volume of DMEM was mixed with viruses for control groups, and then the mixtures were incubated at 37°C for 1 h. Plates were incubated at 37°C in 5% $CO_2$ for 4 to 5 days or until CPE was observed. The effective dilution of sera to 50% endpoint titers (NT50) was calculated using the Karber method.

**RNA extraction and RT-qPCR analysis.** RNA extraction and RT-qPCR analysis were performed as described previously (34). Total RNA from the tissues was extracted using RNAiso Plus according to the manufacturer's instructions (TaKaRa, Japan). Tissue transcriptional cytokine expression in spleens was measured using 2 *Taq* SYBR green qPCR premix (Innovagene, Changsha, China) in a CFX Connect real-time PCR detection system (Bio-Rad) following the manufacturer's recommendations. The RT-qPCR system included 0.4 $\mu$L of cDNA, 0.2 $\mu$L of Primer F (Table 1), 0.2 $\mu$L of Primer R (Table 1), 4.2 $\mu$L of RNase-free double-distilled water (ddH$_2$O), and 5 $\mu$L of 2 *Taq* SYBR green qPCR premix. The primers used in this study were described in our previous studies (35).

**Quantification and statistical analysis.** All data were analyzed using GraphPad Prism 8.0 (La Jolla, CA, USA). Data from growth curves were analyzed with multiple $t$ tests. Data from the survival of duck embryos, ducklings, or mice were assessed based on the log-rank (Mantel-Cox) test. The remaining data analysis was performed using an unpaired $t$ test. All of the statistical significances above are defined by $P$ values of $<0.05$. For animal studies, ducks and mice were randomly allocated to different cages before experiments, and no animals were excluded from analyses. The investigator was blinded to group allocation when the virus was detected from collected tissues.

**Data availability.** The raw data supporting the conclusions of this article will be made available by the authors without undue reservation. To demonstrate the stability of our recombinant virus, we sequenced the virus in mouse and duck embryos as well as in BHK-21 cells to verify (SRA accession numbers SRR22762464, SRR22762463, SRR22762464, SRR22762461, SRR22762460, SRR22762462).

## ACKNOWLEDGMENTS

This work was funded by grants from the China Central and Eastern European countries joint education project (2021092), Sichuan Provincial Department of Science and Technology International Scientific and Technological Innovation Cooperation (2022YFH0026), the China Agricultural Research System (CARS-42-17), and the Program Sichuan Veterinary Medicine and Drug Innovation Group of China Agricultural Research System (SCCXTD-2021-18).

We declare that there are no competing financial interests regarding the publication of this paper.

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
