## [Reviewer comments · Microbiology Spectrum]

Microbiology Spectrum

The attenuation of avian flavivirus by rewiring the leucine and serine codons of its E-NS1 protein towards stop mutation to redirecting virus evolution

Jiaqi Guo, Shun Chen, Yu He, Xiaoli Wang, Andres Merits, Mingshu Wang, Renyong Jia, Dekang Zhu, Mafeng Liu, Xinxin Zhao, Qiao Yang, Ying Wu, Shaqiu Zhang, Juan Huang, Sai Mao, Xumin Ou, Qun Gao, Di Sun, Bin Tian, and Anchun Cheng

Corresponding Author(s): Shun Chen, Sichuan Agricultural University

Review Timeline:

Submission Date:	July 28, 2022
Editorial Decision:	August 23, 2022
Revision Received:	October 29, 2022
Editorial Decision:	November 1, 2022
Revision Received:	November 10, 2022
Accepted:	November 21, 2022

Editor: Peter Pelka

Reviewer(s): Disclosure of reviewer identity is with reference to reviewer comments included in decision letter(s). The following individuals involved in review of your submission have agreed to reveal their identity: ZONGYAN CHEN (Reviewer #1); Scott B. Biering (Reviewer #2)

Transaction Report:

DOI: <https://doi.org/10.1128/spectrum.02921-22>

August 23, 2022

Dr. Shun Chen
Sichuan Agricultural University
Institute of Preventive Veterinary Medicine
No. 211 Huimin Road
Wenjiang District
Chengdu, Sichuan Province 611130
China

Re: Spectrum02921-22 (A trail on the attenuation of avian flavivirus by redirecting their evolution in sequence space)

Dear Dr. Shun Chen:

Thank you for submitting your manuscript to Spectrum. I have received reviews for your manuscript, appended below, and I would request that you address the major concerns of the two reviewers prior to resubmission. In particular, reviewer #2 makes important points regarding sequencing of the generated viruses to show stability and increased frequency of stop codons within genomic RNA.

Link Not Available

Sincerely,

Peter Pelka

Journals Department
Reviewer comments:

Reviewer #1 (Comments for the Author):

L26: What is the CQW1-OTS? It should be briefly described in the abstract.

L76: Please clarify the abbreviate of DTMOV.

L85: "series" should be "serine".

L100: Does CQW1-WT means wild-type CQW1? Please clarify the abbreviate when it appears at the first time.
L180-187: The procedure for mutagenic reagent treatments is confusing, please reedit this paragraph.
L210: The abbreviate of "IC" has been clarified in L81.
L276: In figure 1E, viable viruses are generated at first round, why didn't the CQW1-OTS generate any viable virus after five rounds of treatment?
L346-348: This discussion/conclusion should not be place in the results section.
L365: ...the TMUV strain we used was CQW1...
L383-384: Please reedit this sentence, it's hard to understand.
L384-385: For Influenza A virus and Coxsackie virus B3, both of their natural hosts...
L390-399: The authors proposed that relocation of RNA virus in sequence space to achieve attenuation depend on the sufficient external pressure. So, did the strategy in PCV2 follow this rule?
Figure 1A: What's the meaning of the "T→C"? I did not see any related information in the main body of the manuscript.
Figure 1D/E: Please add the baseline value for the TCID50 method, and this also apply to other figures.

Reviewer #2 (Comments for the Author):

Guo et al., aim to utilize the recently described 1-to-stop method of viral attenuation on flaviviruses to determine if such a method is possible for this viral family with hopes of future development of live-attenuated vaccines against flaviviruses. The authors use duck Tembusu virus (TMUV) as a model flavivirus and mutate the 69 serine and 53 leucine residues within the E-NS1 sequence synonymously such that the amino acid was conserved but that the codon was one nucleotide away from a stop codon, potentially limiting the evolutionary space of the virus. The authors successfully rescue mutant virus which they show replicate comparably with the parental strain in both cell culture and in duck embryos. They then measure whether the virus is attenuated in multiple in vivo models. They find that while there is not significant attenuation in a younger duck infection model, there is statistically significant (although not dramatic) attenuation within an older duck model as well as an intracranial route-mouse model. The authors hypothesize that different selective pressure in these models contribute to the presence or absence of mutant virus attenuation. The data are not dramatic but the trends seem consistent. A major question that must be addressed (and is even pointed out by the authors) is if the mild attenuation observed stems from the proposed model of poor adaptability of the mutant virus from constrained evolutionary space coming from the 1-to-stop approach (as the authors speculate) or else from a broader attenuation owing from mutating many nucleotides within the virus genome which can alter protein expression from alternative codon usage or even other unknown attenuation mechanisms. The authors need to sequence the virus found within in vivo organs to determine stop codon percentage of wt vs. mutant virus as well as look for virus reversion to WT sequences. Without the addition of these critical control experiments it is unknown if the authors approach was successful. Other points to consider:

1. The authors could potentially rule out the possibility of the mutagenesis causing attenuation independently from constraining evolutionary via the addition of a control mutant virus where they mutate serines and leucines to alternative codons that are two nucleotides away from a stop codon as reported previously, is this possible?
2. Line 81, I would not call an intracranial injection infection model of a flavivirus a "perfect model". It's inappropriate to do so. In fact, this IC route of administration may result in too strong of an infection to see attenuation of the mutant virus. Is there an alternative mouse model or route of administration that can be used?
3. For figure 1B and 1C WT and mutant viruses must be compared in the same assay.
4. Figure 2-4 duckling, older ducks, mice, and embryo experiment (and cells for that matter), please sequence mutant and WT viruses during infections to look for reversion mutations. Are your mutations stable? Is it possible to monitor stop codon rates in viral sequences from animal organs as done in Moratorio et al. Nature Microbiology 2017.
5. English should be clarified throughout the manuscript as in some places the meaning of the authors is obscure, for example the title of the manuscript is unclear.
6. Line 39 "But no cure existed until today" needs to be changed to something like but no approved treatments are currently available. As worded it sounds like the authors present a cure.

Staff Comments:

Preparing Revision Guidelines

- Point-by-point responses to the issues raised by the reviewers in a file named "Response to Reviewers," NOT IN YOUR

COVER LETTER.

- Upload a compare copy of the manuscript (without figures) as a "Marked-Up Manuscript" file.
- Each figure must be uploaded as a separate file, and any multipanel figures must be assembled into one file.
- Manuscript: A .DOC version of the revised manuscript
- Figures: Editable, high-resolution, individual figure files are required at revision, TIFF or EPS files are preferred

Please return the manuscript within 60 days; if you cannot complete the modification within this time period, please contact me. If you do not wish to modify the manuscript and prefer to submit it to another journal, please notify me of your decision immediately so that the manuscript may be formally withdrawn from consideration by Microbiology Spectrum.

In the present manuscript, the authors attempted to apply a universal method into developing live attenuated vaccines for flavivirus based on the new strategy - the sequence space, which is an interesting theory. The viral genome defines the location of virus populations in sequence space, and their positions will be changed when the bases of viral genome are adjusted accordingly. By re-editing the codons of leucine and serine of viral ORF, virus genome could be placed on a position which face a detrimental mutational-neighborhood, results in virus genome tend to produce lethal mutations, achieving attenuation. Using TMUV a model, the author demonstrated that there is a precondition for this strategy— sufficient external pressure. Overall, the data is clearly presented, the major findings convincing and the outcomes reasonably well discussed and with the improvements suggested below will make a valuable addition to the literature.

L26: What is the CQW1-OTS? It should be briefly described in the abstract.

L76: Please clarify the abbreviate of DTMUV.

L85: “series” should be “serine”.

L100: Does CQW1-WT means wild-type CQW1? Please clarify the abbreviate when it appears at the first time.

L180-187: The procedure for mutagenic reagent treatments is confusing, please reedit this paragraph.

L210: The abbreviate of “IC” has been clarified in L81.

L276: In figure 1E, viable viruses are generated at first round, why didn't the CQW1-OTS generate any viable virus after five rounds of treatment?

L346-348: This discussion/conclusion should not be place in the results section.

L365: ...the TMUV strain we used was CQW1...

L383-384: Please reedit this sentence, it's hard to understand.

L384-385: For Influenza A virus and Coxsackie virus B3, both of their natural hosts...

L390-399: The authors proposed that relocation of RNA virus in sequence space to achieve attenuation depend on the sufficient external pressure. So, did the strategy in PCV2 follow this rule?

Figure 1A: What's the meaning of the “T→C”? I did not see any related information in the main body of the manuscript.

Figure 1D/E: Please add the baseline value for the TCID50 method, and this also apply to other figures.

Sichuan Agricultural University

College of Veterinary Medicine
Institute of Preventive Veterinary Medicine

No. 211 Huimin Road, Wenjiang District,
Chengdu City, Sichuan Province, China,

October 19, 2022

Dear *Microbiology Spectrum* Editor,

Manuscript Number: **Spectrum02921-22R1**

Title: A trail on the attenuation of avian flavivirus by redirecting their evolution in sequence space

Thank you for the kind letter on August, 26th, 2022. We have revised the manuscript in accordance with every comment of reviewers, and carefully proofread the manuscript again to minimize typographical, grammatical, and bibliographical errors.

Please check our description on revision point-by-point according to the comments as follows.

Reviewer 1

Major points:

1. L26: What is the CQW1-OTS? It should be briefly described in the abstract.

The author's response:

Thanks for the kind comments.

Your suggestions are very helpful to our article, and we have added the CQW1-OTS description to the abstract. The changes are highlighted in Yellow in the revised manuscript (lines 26-27). Please kindly check it.

Sichuan Agricultural University

College of Veterinary Medicine
Institute of Preventive Veterinary Medicine

No. 211 Huimin Road, Wenjiang District,
Chengdu City, Sichuan Province, China,

2. L76: Please clarify the abbreviate of DTMUV.

The author's response:

Thanks for the kind comments.

Your suggestions are very helpful to our article, and we have clarified the abbreviation DTMUV. The changes are highlighted in Yellow in the revised manuscript (line79).

Please kindly check it.

3. L85: "series" should be "serine".

The author's response:

Thanks for the kind comments.

Your suggestions are very helpful to our article, and we have modified it according to the suggestion. The changes are highlighted in Yellow in the revised manuscript (line 88). Please kindly check it.

4. L100: Does CQW1-WT means wild-type CQW1? Please clarify the abbreviate when it appears at the first time.

The author's response:

Sichuan Agricultural University

College of Veterinary Medicine
Institute of Preventive Veterinary Medicine

No. 211 Huimin Road, Wenjiang District,
Chengdu City, Sichuan Province, China,

Thanks for kind comments.

CQW1-WT means wild-type CQW1, and we have added details where they first appeared (lines 26-27), making it more understandable. Please kindly check it.

5. L180-187: The procedure for mutagenic reagent treatments is confusing, please reedit this paragraph.

The author's response:

Thanks for the kind comments.

Your suggestions are very helpful to our article, and we have re-edited this paragraph to make it easier to read. (lines185-191). Please kindly check it.

6. 6. L210: The abbreviate of "IC" has been clarified in L81.

The author's response:

Thanks for the kind comments. We have changed it at line 213. Please kindly check it.

7. L276: In figure 1E, viable viruses are generated at first round, why didn't the CQW1-OTS generate any viable virus after five rounds of treatment?

The author's response:

Sichuan Agricultural University

College of Veterinary Medicine
Institute of Preventive Veterinary Medicine

No. 211 Huimin Road, Wenjiang District,
Chengdu City, Sichuan Province, China,

Thanks for kind comments.

First, because we measured the virus titer of its supernatant using TCID₅₀, there is a possibility that a few viruses are still alive but blew the detection limit after five rounds of treatment.

Second, based on our CQW1-OTS design, the greater and longer the pressure driven by drugs increased its mutation, the greater the number of mutations to the stop codon in the virus population, and the lower the virus titer to be.

We believe this is reason why CQW1-OTS was not detected to produce any live virus after five rounds of treatment.

7. L346-348: This discussion/conclusion should not be place in the results section.

The author's response:

Thanks for the kind comments.

Your suggestions are very helpful to our article, and we have revised this sentence (lines 350-351). Please kindly check it.

8. L365: ...the TMUV strain we used was CQW1...

The author's response:

Sichuan Agricultural University

College of Veterinary Medicine
Institute of Preventive Veterinary Medicine

No. 211 Huimin Road, Wenjiang District,
Chengdu City, Sichuan Province, China,

Thanks for the kind comments.

Your suggestions are very helpful to our article, and we have modified it according to your suggestion (line 368). Please kindly check it.

9. L383-384: Please reedit this sentence, it's hard to understand.

The author's response:

Thanks for the kind comments.

Your suggestions are very helpful to our article, and we have re-edited this paragraph to make it easier to understand. (lines382-389). Please kindly check it.

11. L384-385: For Influenza A virus and Coxsackie virus B3, both of their natural hosts...

The author's response:

Thanks for the kind comments.

Your suggestions are very helpful to our article, and we have modified it according to your suggestion (lines 386-389). Please kindly check it.

12. L390-399: The authors proposed that relocation of RNA virus in sequence space

Sichuan Agricultural University

College of Veterinary Medicine
Institute of Preventive Veterinary Medicine

No. 211 Huimin Road, Wenjiang District,
Chengdu City, Sichuan Province, China,

to achieve attenuation depend on the sufficient external pressure. So, did the strategy in PCV2 follow this rule?

The author's response:

Thanks for kind comments.

Yes, we assumed that the policy in PCV2 follows this rule. There are several main reasons for this. Because the TMUV virus and PCV2 viruses presented in this paper use the same attenuation method (1), it is applicable. Deep sequencing of PCV2 reveals that it develops mutations as a result of selection pressure, with the majority of these being deadly (2).

- (1) Moratorio G, Henningson R, Barbezange C, Carrau L, Borderia AV, Blanc H, Beaucourt S, Poirier EZ, Vallet T, Boussier J, Mounce BC, Fontes M, Vignuzzi M. 2017. Attenuation of RNA viruses by redirecting their evolution in sequence space. *Nat Microbiol* 2:17088.
- (2) Ramamoorthy S. 2021. Harnessing the Genetic Plasticity of Porcine Circovirus Type 2 to Target Suicidal Replication. *Viruses* 13.

13. Figure 1A: What's the meaning of the "T→C"? I did not see any related information in the main body of the manuscript.

The author's response:

Thanks for kind comments. We have explained the meaning of "T→C" in our constructs in the Materials and methods section of the revised manuscript, changes are

Sichuan Agricultural University

College of Veterinary Medicine
Institute of Preventive Veterinary Medicine

No. 211 Huimin Road, Wenjiang District,
Chengdu City, Sichuan Province, China,

highlighted in yellow (lines 116-119). Please kindly check it.

14. Figure 1D/E: Please add the baseline value for the TCID₅₀ method, and this also apply to other figures.

The author's response:

Thanks for kind comments. We have modified the Figure 1D/E, Figure 2F and Figure 4C, making them clear and more understandable. Please kindly check it.

Reviewer 2

Major points:

1. The authors could potentially rule out the possibility of the mutagenesis causing attenuation independently from constraining evolutionary via the addition of a control mutant virus where they mutate serines and leucines to alternative codons that are two nucleotides away from a stop codon as reported previously, is this possible?

The author's response:

Thanks for the comments. Yes, we agree with you.

Sichuan Agricultural University

College of Veterinary Medicine
Institute of Preventive Veterinary Medicine

No. 211 Huimin Road, Wenjiang District,
Chengdu City, Sichuan Province, China,

2. Line 81, I would not call an intracranial injection infection model of a flavivirus a "perfect model". It's inappropriate to do so. In fact, this IC route of administration may result in too strong of an infection to see attenuation of the mutant virus. Is there an alternative mouse model or route of administration that can be used?

The author's response:

Thanks for kind comments.

Yes, this statement may be poorly considered, and we have made changes to the sub-description (lines 84-85). Dawei Yan et al.(3, 4 in Chinese articles) found that some TMUV strains could not establish effective systemic infection by nasal drip. In previous studies we also tried this route by intraperitoneal injection, but shown low viral tissue loads (data not published). So relatively speaking, the most effective route of infection with TMUV in for 3-week-old Kunming mice is intracranial injection.

(3) 闫大为. 2015. 坦布苏病毒 MM1775 株反向遗传操作系统的建立及拯救病毒的生物学特性研究中国农业科学院.

(4) 闫大为. 2018. 坦布苏病毒对鸭及小鼠致病的分子基础研究中国农业科学院.

3. For figure 1B and 1C WT and mutant viruses must be compared in the same assay.

Sichuan Agricultural University

College of Veterinary Medicine
Institute of Preventive Veterinary Medicine

No. 211 Huimin Road, Wenjiang District,
Chengdu City, Sichuan Province, China,

The author's response:

Thanks for kind comments.

Your suggestions are very helpful to our article, and we have re-edited the Figure according to the suggestion. Please kindly check it.

4. Figure 2-4 duckling, older ducks, mice, and embryo experiment (and cells for that matter), please sequence mutant and WT viruses during infections to look for reversion mutations. Are your mutations stable? Is it possible to monitor stop codon rates in viral sequences from animal organs as done in Moratorio et al. *Nature Microbiology* 2017.

The author's response:

Thanks for kind comments.

We sequenced the recombinant virus of infected animal tissues (mouse brain and embryo) as well as cell samples, and no reversion mutations were observed (please kindly check the sequence matching results in the **Supplemental Material**, named “embryo CQW1-WT, embryo CQW1-OTS, cell CQW1-OTS, cell CQW1-WT, mouse brain CQW1-OTS and mouse brain CQW1-WT”).

5. English should be clarified throughout the manuscript as in some places the

Sichuan Agricultural University

College of Veterinary Medicine
Institute of Preventive Veterinary Medicine

No. 211 Huimin Road, Wenjiang District,
Chengdu City, Sichuan Province, China,

meaning of the authors is obscure, for example the title of the manuscript is unclear.

The author's response:

Thanks for kindly comments, we have carefully proofread the manuscript to minimize errors. Meanwhile, the revised manuscript has been edited twice by a professional service of English correction from American Journal Experts (Certificate Verification Key: C1DA-C8F4-932B-7B63-2A58).

AJE Editing Certificate

This document certifies that the manuscript

The attenuation of avian flavivirus by rewiring the leucine and serine codons of its E-NS1 protein towards stop mutation to redirecting virus evolution

prepared by the authors

Jiaqi Guo, Yu He, Xiaoli Wang, Anders Mertis, Mingshu Wang, Renyong Jia, Dekang Zhu, Mafeng Liu, Xinxin Zhao, Qiao Yang, Ying Wu, Shaiqi Zhang, Juan Huang, Sai Mao, Xumin Ou, Qun Gao, Di Sun, Bin Tian, Anchun Cheng and Shun Chen

was edited for proper English language, grammar, punctuation, spelling, and overall style by one or more of the highly qualified native English speaking editors at AJE.

This certificate was issued on **October 14, 2022** and may be verified on the AJE website using the verification code **C1DA-C8F4-932B-7B63-2A58**.

Neither the research content nor the authors' intentions were altered in any way during the editing process. Documents receiving this certification should be English-ready for publication; however, the author has the ability to accept or reject our suggestions and changes. To verify the final AJE edited version, please visit our verification page at aje.com/certificate. If you have any questions or concerns about this edited document, please contact AJE at support@aje.com.

AJE provides a range of editing, translation, and manuscript services for researchers and publishers around the world. For more information about our company, services, and partner discounts, please visit aje.com.

And we changed the title into: The attenuation of avian flavivirus by rewiring the leucine and serine codons of its E-NS1 protein towards stop mutation redirecting virus

Sichuan Agricultural University

College of Veterinary Medicine
Institute of Preventive Veterinary Medicine

No. 211 Huimin Road, Wenjiang District,
Chengdu City, Sichuan Province, China,

evolution. Please kindly check it.

6. Line 39 "But no cure existed until today" needs to be changed to something like but approved treatments are currently available. As worded it sounds like the authors present a cure.

The author's response:

Thanks for the kind comments.

Your suggestions are very helpful to our article, and We have revised this paragraph to "Flaviviruses are medically important arboviruses that threaten public health, but no approved treatments are currently available" as suggested (lines 40-41). Please kindly check it.

Finally, we greatly appreciate all helpful comments and suggestions in our manuscript, because they are valuable in improving the quality of our manuscript.

Please do not hesitate to contact with us if you have any question. We are looking forward to hearing from you.

Sincerely,

Sichuan Agricultural University

College of Veterinary Medicine
Institute of Preventive Veterinary Medicine

No. 211 Huimin Road, Wenjiang District,
Chengdu City, Sichuan Province, China,

Shun Chen, PhD, Professor

Institute of Preventive Veterinary Medicine

College of Veterinary Medicine, Sichuan Agricultural University

Mobile: +86-189-8082-5808

Email: shunchen@sicau.edu.cn; sophia_cs@163.com.

November 1, 2022

Dr. Shun Chen
Sichuan Agricultural University
Institute of Preventive Veterinary Medicine
No. 211 Huimin Road
Wenjiang District
Chengdu, Sichuan Province 611130
China

Re: Spectrum02921-22R1 (The attenuation of avian flavivirus by rewiring the leucine and serine codons of its E-NS1 protein towards stop mutation to redirecting virus evolution)

Dear Dr. Shun Chen:

Thank you for submitting a revised manuscript, upon reviewing the response to the prior comments I find that most are satisfactory. However, the response to reviewer #2 major comment #1 is not and I would request clarification before I can make a decision on the manuscript. Specifically, I am referring to this comment:

The authors could potentially rule out the possibility of the mutagenesis causing attenuation independently from constraining evolutionary via the addition of a control mutant virus where they mutate serines and leucines to alternative codons that are two nucleotides away from a stop codon as reported previously, is this possible?

Link Not Available

Sincerely,

Peter Pelka

Journals Department
Reviewer comments:

Staff Comments:

Preparing Revision Guidelines

Please return the manuscript within 60 days; if you cannot complete the modification within this time period, please contact me. If you do not wish to modify the manuscript and prefer to submit it to another journal, please notify me of your decision immediately so that the manuscript may be formally withdrawn from consideration by Microbiology Spectrum.

Sichuan Agricultural University

College of Veterinary Medicine
Institute of Preventive Veterinary Medicine

No. 211 Huimin Road, Wenjiang District,
Chengdu City, Sichuan Province, China,

October 19, 2022

Dear *Microbiology Spectrum* Editor,

Manuscript Number: **Spectrum02921-22R1**

Title: A trail on the attenuation of avian flavivirus by redirecting their evolution in sequence space

Thank you for the kind letter on August, 26th, 2022. We have revised the manuscript in accordance with every comment of reviewers, and carefully proofread the manuscript again to minimize typographical, grammatical, and bibliographical errors.

Please check our description on revision point-by-point according to the comments as follows.

Reviewer 1

Major points:

1. L26: What is the CQW1-OTS? It should be briefly described in the abstract.

The author's response:

Thanks for the kind comments.

Your suggestions are very helpful to our article, and we have added the CQW1-OTS description to the abstract. The changes are highlighted in Yellow in the revised manuscript (lines 26-27). Please kindly check it.

Sichuan Agricultural University

College of Veterinary Medicine
Institute of Preventive Veterinary Medicine

No. 211 Huimin Road, Wenjiang District,
Chengdu City, Sichuan Province, China,

2. L76: Please clarify the abbreviate of DTMUV.

The author's response:

Thanks for the kind comments.

Your suggestions are very helpful to our article, and we have clarified the abbreviation DTMUV. The changes are highlighted in Yellow in the revised manuscript (line79).

Please kindly check it.

3. L85: "series" should be "serine".

The author's response:

Thanks for the kind comments.

Your suggestions are very helpful to our article, and we have modified it according to the suggestion. The changes are highlighted in Yellow in the revised manuscript (line 88). Please kindly check it.

4. L100: Does CQW1-WT means wild-type CQW1? Please clarify the abbreviate when it appears at the first time.

The author's response:

Sichuan Agricultural University

College of Veterinary Medicine
Institute of Preventive Veterinary Medicine

No. 211 Huimin Road, Wenjiang District,
Chengdu City, Sichuan Province, China,

Thanks for kind comments.

CQW1-WT means wild-type CQW1, and we have added details where they first appeared (lines 26-27), making it more understandable. Please kindly check it.

5. L180-187: The procedure for mutagenic reagent treatments is confusing, please reedit this paragraph.

The author's response:

Thanks for the kind comments.

Your suggestions are very helpful to our article, and we have re-edited this paragraph to make it easier to read. (lines185-191). Please kindly check it.

6. 6. L210: The abbreviate of "IC" has been clarified in L81.

The author's response:

Thanks for the kind comments. We have changed it at line 213. Please kindly check it.

7. L276: In figure 1E, viable viruses are generated at first round, why didn't the CQW1-OTS generate any viable virus after five rounds of treatment?

The author's response:

Sichuan Agricultural University

College of Veterinary Medicine
Institute of Preventive Veterinary Medicine

No. 211 Huimin Road, Wenjiang District,
Chengdu City, Sichuan Province, China,

Thanks for kind comments.

First, because we measured the virus titer of its supernatant using TCID₅₀, there is a possibility that a few viruses are still alive but blew the detection limit after five rounds of treatment.

Second, based on our CQW1-OTS design, the greater and longer the pressure driven by drugs increased its mutation, the greater the number of mutations to the stop codon in the virus population, and the lower the virus titer to be.

We believe this is reason why CQW1-OTS was not detected to produce any live virus after five rounds of treatment.

7. L346-348: This discussion/conclusion should not be place in the results section.

The author's response:

Thanks for the kind comments.

Your suggestions are very helpful to our article, and we have revised this sentence (lines 350-351). Please kindly check it.

8. L365: ...the TMUV strain we used was CQW1...

The author's response:

Sichuan Agricultural University

College of Veterinary Medicine
Institute of Preventive Veterinary Medicine

No. 211 Huimin Road, Wenjiang District,
Chengdu City, Sichuan Province, China,

Thanks for the kind comments.

Your suggestions are very helpful to our article, and we have modified it according to your suggestion (line 368). Please kindly check it.

9. L383-384: Please reedit this sentence, it's hard to understand.

The author's response:

Thanks for the kind comments.

Your suggestions are very helpful to our article, and we have re-edited this paragraph to make it easier to understand. (lines382-389). Please kindly check it.

11. L384-385: For Influenza A virus and Coxsackie virus B3, both of their natural hosts...

The author's response:

Thanks for the kind comments.

Your suggestions are very helpful to our article, and we have modified it according to your suggestion (lines 386-389). Please kindly check it.

12. L390-399: The authors proposed that relocation of RNA virus in sequence space

Sichuan Agricultural University

College of Veterinary Medicine
Institute of Preventive Veterinary Medicine

No. 211 Huimin Road, Wenjiang District,
Chengdu City, Sichuan Province, China,

to achieve attenuation depend on the sufficient external pressure. So, did the strategy in PCV2 follow this rule?

The author's response:

Thanks for kind comments.

Yes, we assumed that the policy in PCV2 follows this rule. There are several main reasons for this. Because the TMUV virus and PCV2 viruses presented in this paper use the same attenuation method (1), it is applicable. Deep sequencing of PCV2 reveals that it develops mutations as a result of selection pressure, with the majority of these being deadly (2).

- (1) Moratorio G, Henningson R, Barbezange C, Carrau L, Borderia AV, Blanc H, Beaucourt S, Poirier EZ, Vallet T, Boussier J, Mounce BC, Fontes M, Vignuzzi M. 2017. Attenuation of RNA viruses by redirecting their evolution in sequence space. *Nat Microbiol* 2:17088.
- (2) Ramamoorthy S. 2021. Harnessing the Genetic Plasticity of Porcine Circovirus Type 2 to Target Suicidal Replication. *Viruses* 13.

13. Figure 1A: What's the meaning of the "T→C"? I did not see any related information in the main body of the manuscript.

The author's response:

Thanks for kind comments. We have explained the meaning of "T→C" in our constructs in the Materials and methods section of the revised manuscript, changes are

Sichuan Agricultural University

College of Veterinary Medicine
Institute of Preventive Veterinary Medicine

No. 211 Huimin Road, Wenjiang District,
Chengdu City, Sichuan Province, China,

highlighted in yellow (lines 116-119). Please kindly check it.

14. Figure 1D/E: Please add the baseline value for the TCID₅₀ method, and this also apply to other figures.

The author's response:

Thanks for kind comments. We have modified the Figure 1D/E, Figure 2F and Figure 4C, making them clear and more understandable. Please kindly check it.

Reviewer 2

Major points:

1. The authors could potentially rule out the possibility of the mutagenesis causing attenuation independently from constraining evolutionary via the addition of a control mutant virus where they mutate serines and leucines to alternative codons that are two nucleotides away from a stop codon as reported previously, is this possible?

The author's response:

Thanks for the comments.

1. We don't quite understand the reviewer's comment.

Sichuan Agricultural University

College of Veterinary Medicine
Institute of Preventive Veterinary Medicine

No. 211 Huimin Road, Wenjiang District,
Chengdu City, Sichuan Province, China,

2. If the point comment is for us to add a new control strain that is two bases away from the stop codon, there is no way to do it at the moment because the protocol is not designed for that strain. Also, we did not consider it significant to construct a strain where the distance terminator is a synonymous mutation of two nucleotides to serve as a control. Because this did not involve amino acid changes nor did it reach the level of de-optimization, it was not considered.

2. Line 81, I would not call an intracranial injection infection model of a flavivirus a "perfect model". It's inappropriate to do so. In fact, this IC route of administration may result in too strong of an infection to see attenuation of the mutant virus. Is there an alternative mouse model or route of administration that can be used?

The author's response:

Thanks for kind comments.

Yes, this statement may be poorly considered, and we have made changes to the sub-description (lines 84-85). Dawei Yan et al.(3, 4 in Chinese articles) found that some TMUV strains could not establish effective systemic infection by nasal drip. In previous studies we also tried this route by intraperitoneal injection, but shown low viral tissue loads (data not published). So relatively speaking, the most effective route

Sichuan Agricultural University

College of Veterinary Medicine
Institute of Preventive Veterinary Medicine

No. 211 Huimin Road, Wenjiang District,
Chengdu City, Sichuan Province, China,

of infection with TMUV in for 3-week-old Kunming mice is intracranial injection.

(3) 闫大为. 2015. 坦布苏病毒 MM1775 株反向遗传操作系统的建立及拯救病毒的生物学特性

研究中国农业科学院.

(4) 闫大为. 2018. 坦布苏病毒对鸭及小鼠致病的分子基础研究中国农业科学院.

3. For figure 1B and 1C WT and mutant viruses must be compared in the same assay.

The author's response:

Thanks for kind comments.

Your suggestions are very helpful to our article, and we have re-edited the Figure according to the suggestion. Please kindly check it.

4. Figure 2-4 duckling, older ducks, mice, and embryo experiment (and cells for that matter), please sequence mutant and WT viruses during infections to look for reversion mutations. Are your mutations stable? Is it possible to monitor stop codon rates in viral sequences from animal organs as done in Moratorio et al. Nature Microbiology 2017.

The author's response:

Sichuan Agricultural University

College of Veterinary Medicine
Institute of Preventive Veterinary Medicine

No. 211 Huimin Road, Wenjiang District,
Chengdu City, Sichuan Province, China,

Thanks for kind comments.

We sequenced the recombinant virus of infected animal tissues (mouse brain and embryo) as well as cell samples, and no reversion mutations were observed (please kindly check the sequence matching results in the **Miscellaneous File not for Publication**, named “embryo CQW1-WT, embryo CQW1-OTS, cell CQW1-OTS, cell CQW1-WT, mouse brain CQW1-OTS and mouse brain CQW1-WT”, and all the data is just intended for use by the reviewers).

5. English should be clarified throughout the manuscript as in some places the meaning of the authors is obscure, for example the title of the manuscript is unclear.

The author's response:

Thanks for kindly comments, we have carefully proofread the manuscript to minimize errors. Meanwhile, the revised manuscript has been edited twice by a professional service of English correction from American Journal Experts (Certificate Verification Key: C1DA-C8F4-932B-7B63-2A58).

Sichuan Agricultural University

College of Veterinary Medicine
Institute of Preventive Veterinary Medicine

No. 211 Huimin Road, Wenjiang District,
Chengdu City, Sichuan Province, China,

Editing Certificate

This document certifies that the manuscript

The attenuation of avian flavivirus by rewiring the leucine and serine codons of its E-NS1 protein towards stop mutation to redirecting virus evolution

prepared by the authors

Jiaqi Guo, Yu He, Xiaoli Wang, Anders Mertis, Mingshu Wang, Renyong Jia, Dekang Zhu, Mafeng Liu, Xinxin Zhao, Gao Yang, Ying Wu, Shaqiu Zhang, Juan Huang, Sai Mao, Xumin Ou, Qun Gao, Di Sun, Bin Tian, Anchun Cheng and Shun Chen

was edited for proper English language, grammar, punctuation, spelling, and overall style by one or more of the highly qualified native English speaking editors at AJE.

This certificate was issued on **October 14, 2022** and may be verified on the AJE website using the verification code **C1DA-C8F4-932B-7B63-2A58**.

Neither the research content nor the authors' intentions were altered in any way during the editing process. Documents receiving this certification should be English-ready for publication; however, the author has the ability to accept or reject our suggestions and changes. To verify the final AJE edited version, please visit our verification page at aje.com/certificate. If you have any questions or concerns about this edited document, please contact AJE at support@aje.com.

AJE provides a range of editing, translation, and manuscript services for researchers and publishers around the world. For more information about our company, services, and partner discounts, please visit aje.com.

And we changed the title into: The attenuation of avian flavivirus by rewiring the leucine and serine codons of its E-NS1 protein towards stop mutation redirecting virus evolution. Please kindly check it.

6. Line 39 "But no cure existed until today" needs to be changed to something like but no approved treatments are currently available. As worded it sounds like the authors present a cure.

The author's response:

Sichuan Agricultural University

College of Veterinary Medicine
Institute of Preventive Veterinary Medicine

No. 211 Huimin Road, Wenjiang District,
Chengdu City, Sichuan Province, China,

Thanks for the kind comments.

Your suggestions are very helpful to our article, and We have revised this paragraph to "Flaviviruses are medically important arboviruses that threaten public health, but no approved treatments are currently available" as suggested (lines 40-41). Please kindly check it.

Finally, we greatly appreciate all helpful comments and suggestions in our manuscript, because they are valuable in improving the quality of our manuscript.

Please do not hesitate to contact with us if you have any question. We are looking forward to hearing from you.

Sincerely,

Shun Chen, PhD, Professor

Institute of Preventive Veterinary Medicine

College of Veterinary Medicine, Sichuan Agricultural University

Mobile: +86-189-8082-5808

Email: shunchen@sicau.edu.cn; sophia_cs@163.com.

November 21, 2022

Dr. Shun Chen
Sichuan Agricultural University
Institute of Preventive Veterinary Medicine
No. 211 Huimin Road
Wenjiang District
Chengdu, Sichuan Province 611130
China

Re: Spectrum02921-22R2 (The attenuation of avian flavivirus by rewiring the leucine and serine codons of its E-NS1 protein towards stop mutation to redirecting virus evolution)

Dear Dr. Shun Chen:

Your manuscript has been accepted, and I am forwarding it to the ASM Journals Department for publication. You will be notified when your proofs are ready to be viewed.

Sincerely,

Peter Pelka
Editor, Microbiology Spectrum
